# One-Tip enables comprehensive proteome coverage in minimal cells and single zygotes

Zilu Ye [1,2,7] ✉, Pierre Sabatier [2,3,7], Javier Martin-Gonzalez [4], Akihiro Eguchi [2], Maico Lechner [2], Ole Østergaard [2], Jingsheng Xie[1], Yuan Guo[5], Lesley Schultz[5], Rafaela Truffer[5], Dorte B. Bekker-Jensen[6], Nicolai Bache[6] & Jesper V. Olsen [2] ✉

Mass spectrometry (MS)-based proteomics workflows typically involve complex, multi-step processes, presenting challenges with sample losses, reproducibility, requiring substantial time and financial investments, and specialized skills. Here we introduce One-Tip, a proteomics methodology that seamlessly integrates efficient, one-pot sample preparation with precise, narrow-window data-independent acquisition (nDIA) analysis. One-Tip substantially simplifies sample processing, enabling the reproducible identification of >9000 proteins from ~1000 HeLa cells. The versatility of One-Tip is highlighted by nDIA identification of ~6000 proteins in single cells from early mouse embryos. Additionally, the study incorporates the Uno Single Cell Dispenser™, demonstrating the capability of One-Tip in single-cell proteomics with >3000 proteins identified per HeLa cell. We also extend One-Tip workflow to analysis of extracellular vesicles (EVs) extracted from blood plasma, demonstrating its high sensitivity by identifying >3000 proteins from 16 ng EV preparation. One-Tip expands capabilities of proteomics, offering greater depth and throughput across a range of sample types.

Conventional bottom-up proteomics workflows are based on multi-step sample-handling processes, incorporating bulk cell lysis and protein extraction from millions of cells, reduction, and alkylation of cysteines, protein digestion, sample cleanup and concentration, followed by optional offline peptide fractionation prior to online liquid chromatography tandem mass spectrometry (LC-MS/MS) acquisition[1]. The different steps introduce several challenges, including potential sample contamination and losses leading to reproducibility problems, considerable time and cost investments, and the necessity for specialized expertise and training. Consequentially, these barriers often discourage biomedical scientists from undertaking MS-based proteomics experiments.

To address this, several one-pot workflows have emerged, streamlining the sample preparation process for both bulk proteomics[2-4] and single-cell proteomics (SCP)[5-8]. These workflows aim to streamline the sample preparation process while maintaining the depth and accuracy of proteomic analysis. For instance, Kulak et al. introduced an innovative in-StageTip protocol[2], which simplifies proteome-sample preparation down to just five pipetting steps. This method is particularly convenient as it can be easily adapted to a 96-well format, making it suitable for high-throughput applications. The one-pot concept has gained widespread acceptance in the SCP community due to its efficiency and simplicity. Recently, Johnston et al. proposed a sample preparation method that integrates cell lysis,

[1]State Key Laboratory of Common Mechanism Research for Major Diseases, Suzhou Institute of Systems Medicine, Chinese Academy of Medical Sciences & Peking Union Medical College, Suzhou, China. [2]Novo Nordisk Foundation Center for Protein Research, University of Copenhagen, Copenhagen, Denmark. [3]Department of Surgical Sciences, Uppsala University, Uppsala, Sweden. [4]Core Facility for Transgenic Mice, Department of Experimental Medicine, University of Copenhagen, Copenhagen, Denmark. [5]Tecan Group Ltd., Männedorf, Switzerland. [6]Evosep Biosystems, Odense, Denmark. [7]These authors contributed equally: Zilu Ye, Pierre Sabatier. ✉e-mail: yzl@ism.pumc.edu.cn; jesper.olsen@cpr.ku.dk

protein denaturation, and digestion into a single hour-long process[5]. Matzinger et al. presented a comprehensive workflow encompassing improved strategies for all stages of sample preparation, standardized for 384-well plates[6].

Despite these advancements, challenges remain. In bulk proteomics, methods still necessitate substantial sample input, separate cell lysis, multiple liquid handling steps, sample transfers, and extended LC-MS/MS analysis times to achieve the desired proteome depth. These requirements often lead to limited throughput, reproducibility, and applicability, and may not fully capture the proteome depth. One critical limitation in the SCP methods is the lack of a sample cleanup step, which risks compromising the sample quality and hence longevity of LC-MS systems. Recognizing this gap, our study introduces the One-Tip workflow, which is meticulously designed to overcome the prevalent issues in bulk proteomics. One-Tip is primarily tailored to achieve comprehensive proteome depth with high quantitative accuracy and precision, even with a low number of cells. Moreover, we demonstrate the versatility of One-Tip through a wide array of applications, including the analysis of single early embryonic cells and human plasma extracellular vesicle samples. Although not specifically created for SCP, we also show that it is feasible to apply it directly to SCP using the Uno Single Cell dispenser. This innovative and simple workflow promises to transform proteomic sample preparation, offering both efficiency and depth in proteomic analysis.

## Results

### One-Tip epitomizes the most streamlined workflow for proteomics

The One-Tip methodology only requires two pipetting steps, utilizing a commercially available Evotip™: one for the cell lysis and proteolytic digestion buffer, and the other for the cell suspension in PBS (Fig. 1a). After an hour-long incubation in a water-filled Evotip box with standard Evotip preparation, the One-Tip samples are ready for LC-MS/MS analysis. The combined cell lysis, protein extraction, and endoproteinase Lys-C/trypsin digestion master mix buffer includes an MS-compatible surfactant, n-Dodecyl-β-D-Maltoside (DDM), which is a water-soluble non-ionic detergent effective in cell membrane lysis and solubilization of proteins without denaturation.

Conceptually, One-Tip represents the simplest workflow for proteomics, as it necessitates no further sample handling and directly integrates sample preparation with the LC-MS analysis without any sample losses and variation. Low cell input numbers result in high proteome coverage and circumvents the need for sonication and additional buffer exchanges. This results in a total sample preparation time of approximately 70 min, including a digestion time of 60 min. Moreover, the One-Tip workflow is not limited in throughput; even manual handling can efficiently prepare thousands of samples daily by using multi-channel pipetting and is also easily automated on liquid handling robots. Lastly, we couple the One-Tip workflow with a top-of-the-line LC-MS/MS system to guarantee superior analytical performance. This is based on the highly sensitive Whisper LC method on the Evosep One LC and our newly developed narrow-window Data Independent Acquisition (nDIA) method on an Orbitrap Astral mass spectrometer[9].

In the initial experiments, we evaluated different ratios of master mix to HeLa single cell suspension in PBS, testing combinations such as 2 μl master mix + 2 μl cells, 4 μl + 4 μl, and 8 μl + 8 μl, using the previous generation Orbitrap MS platform[10], the Orbitrap Exploris 480 (Fig. 1b). The cell concentration was estimated at 100 cells/μl. Our findings revealed that even the smallest ratio, 2 + 2, yielded a notable 3172 protein group (hereafter referred to as protein) identifications. This protein coverage is comparable to the number of proteins identified in 10 ng HeLa samples prepared using the protein aggregation capture (PAC) protocol[11], a standard reference. Notably, the performance difference observed between the Orbitrap Exploris 480 and the Orbitrap Astral when analyzing One-Tip samples mirrored the differences seen with standard HeLa samples. This consistency indicates that our One-Tip protocol exhibits no bias towards specific mass spectrometry platforms.

To further test the sensitivity and reproducibility of the One-Tip methodology, we deployed it on the Orbitrap Astral using nDIA analysis with increasing numbers of HeLa cells, analyzing 20, 100, 200,

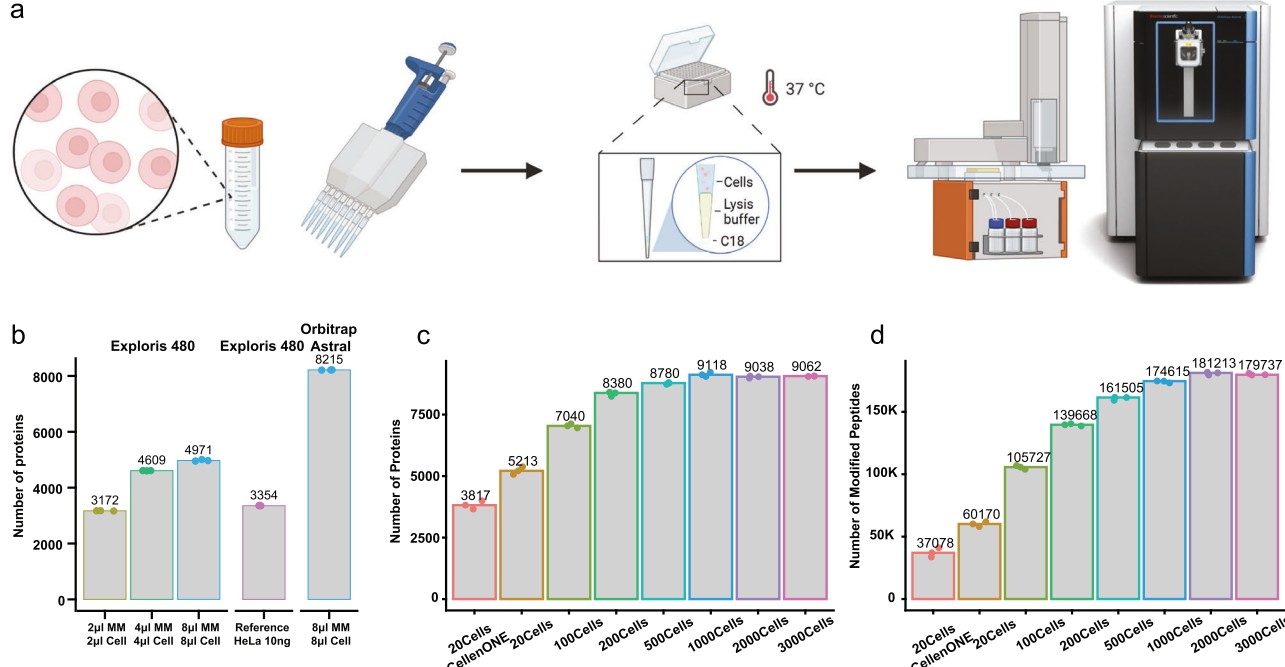

**Fig. 1 | One-tip achieves near-complete proteome depth in single-shot analysis.** **a** Graphic depiction of One-Tip workflow. **b** Number of proteins identified with different volumes of the master mix and cells in PBS. **c**, **d** Number of proteins (**c**) and peptides (**d**) identified in different number of cells. Digestion time was 2 h in these samples. Source data are provided as a Source Data file.

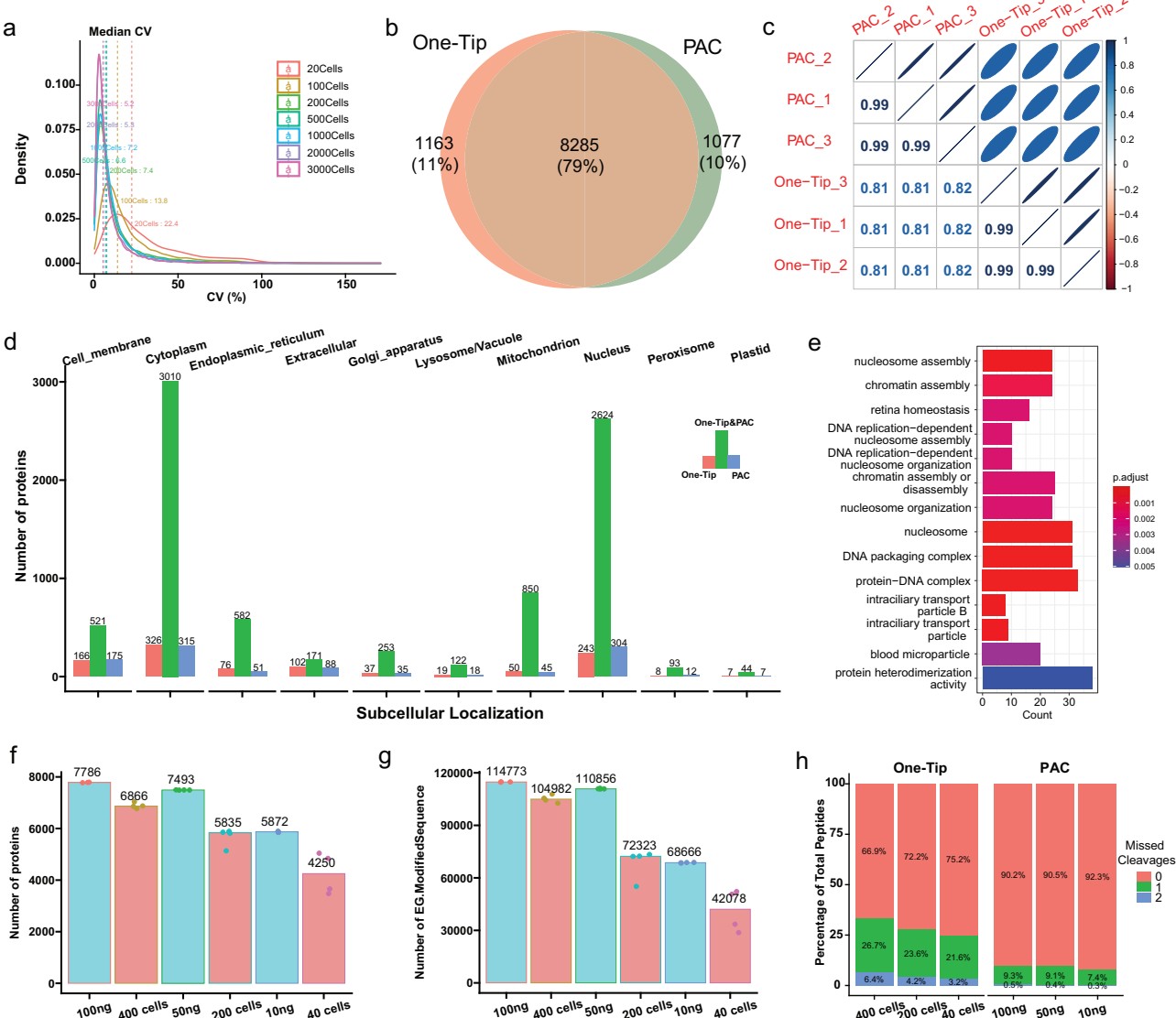

**Fig. 2 | One-Tip shows precise quantification and comparable proteome coverage with PAC. a** Distribution of coefficient of variance between triplicates in different number of cells. **b** Overlap of proteins identified in One-Tip (1000 cells) and PAC (1000 ng) workflows. **c** Correlation of protein abundances between One-Tip and PAC samples. **d** Subcellular localization of proteins identified in One-Tip and PAC samples. **e** GO over-representation enrichment of proteins identified only in One-Tip samples. P values were calculated using a one-sided Fisher's Exact Test and subsequently adjusted using the Benjamini-Hochberg (BH) correction method. **f, g** Number of proteins (**f**) and peptides (**g**) identified in different number of cells using One-Tip and different peptide loading amounts using PAC. **h** Percentage of missed cleavages in One-Tip and PAC samples. Source data are provided as a Source Data file.

500, 1000, 2000, and 3000 cells, respectively. Note these cell quantities were based on cell counting estimations and therefore are approximations. And the volume of 5 μl master mix in this experiment might not be optimal for the different number of cells. One-Tip exhibited an impressive proteome depth for single-shot analysis in half-an-hour LC-MS/MS runs, identifying over 5000 protein groups and over 46,000 modified peptides (hereafter referred to as peptides) from approximately 20 cells (Fig. 1c). This performance surpasses the proteome coverage currently achieved by the isolation of precisely 20 cells using the CellenONE, a dedicated single-cell proteomics preparation system, which is likely related to sample losses in a 96 well plate and transfer to the LC-MS system. Furthermore, a starting quantity of ~1000 cells nearly covered the complete proteome, with over 9000 proteins and 170,000 peptides identified (Fig. 1c, d). We subsequently evaluated the combined endoproteinase Lys-C and trypsin-digestion efficiency by varying the digestion time from 1 to 4 h. Even a proteolytic digestion time as short as 1 h was sufficient to

effectively digest the entire proteome, resulting in high sequence coverage with a missed cleavage rate of less than 25% in 20 cells and approximately 30% for larger cell quantities (Supplementary Figs. 1–3).

### One-Tip shows precise quantification and comparable proteome coverage with PAC using 1000 cells

One-Tip demonstrated high reproducibility and quantitative precision, with a remarkable coefficient of variance of 5–8% in workflow replica analyzes with a Pearson correlation exceeding 0.99 for maxLFQ[12]-based protein quantitation in samples with more than 100 cells (Fig. 2a, Supplementary Fig. 4). Despite its simple and swift workflow, the analytical performance of One-Tip with one thousand cells is comparable to that of bulk proteomics samples prepared using a more sophisticated method, Protein Aggregation Capture (PAC)[11], when combined with our narrow-window DIA method on the Orbitrap Astral mass spectrometer. Compared to 1000-ng PAC-digested HeLa lysate analyzed by LC-MS/MS in >1 h, the One-Tip method applied to 1000-

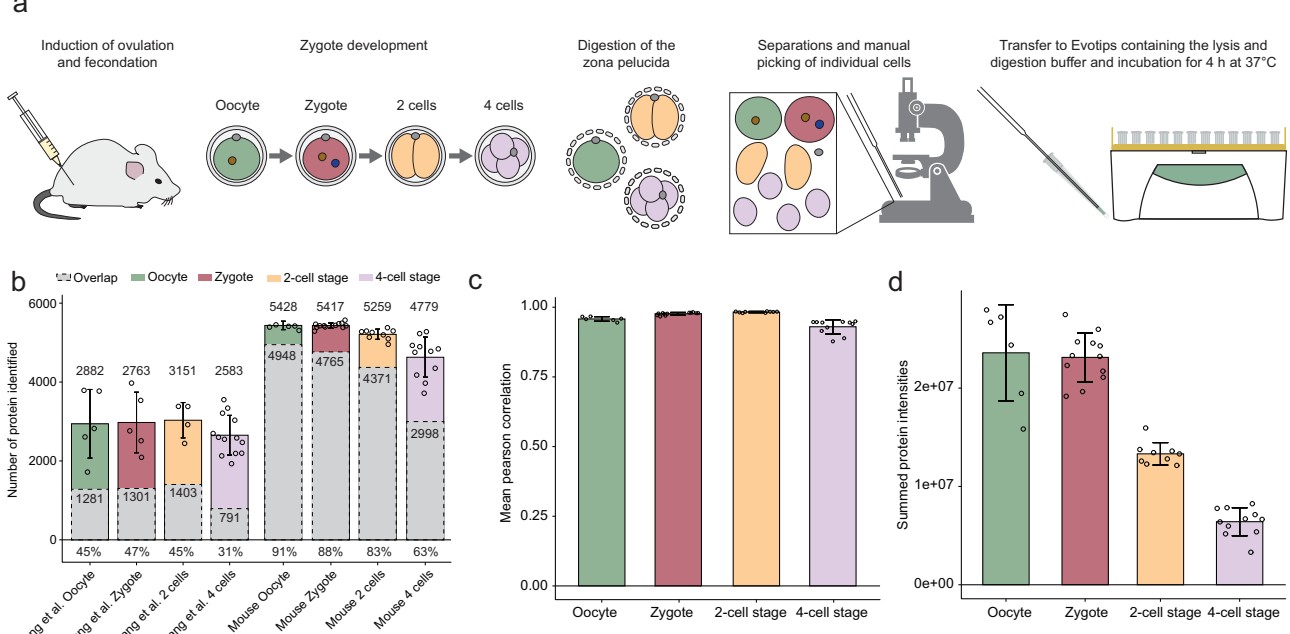

**Fig. 3 | One-Tip and LC-MS analysis of the mouse pre-implantation embryo until the 4-cell stage. a** Graphic depiction of the workflow using One-Tip for mouse pre-implantation embryos. **b** Number of proteins with quantified values in the study from Dang et al. and our study, and protein overlap between samples (in grey) from the same development stages. The mean number of proteins for each condition, the overlap and the mean percentage of the overlap compared to each sample are indicated. **c** Mean correlation of normalized protein abundances between samples within each sample groups (oocyte, zygote, 2-cell stage, 4-cell stage). **d** Summed protein intensities for each group of samples. Error bars represent ± the standard deviation of the mean. Number of biological replicates: $n = 6$ for oocyte, $n = 12$ for zygote, $n = 9$ for 2-cell stage and $n = 11$ for 4-cell stage. Source data are provided as a Source Data file.

cell samples in a half-an-hour LC-MS/MS run identified a similar number of proteins with substantial overlap between them and very high quantification precision (Fig. 2b, c). The lower Pearson correlation of ~0.8 in protein quantities between PAC and One-Tip samples is likely caused by the difference in cysteine-alkylation and missed-cleavage rates (Supplementary Fig. 5). Interestingly, both the PAC and One-Tip methods identified slightly more than 1000 unique proteins, but exhibiting no considerable bias in protein ontologies towards specific subcellular localizations (Fig. 2d), and a slight preference for particular biological functions such as nucleosome assembly in proteins identified only in One-Tip samples (Fig. 2e).

Lastly, to assess the loss of material during One-Tip preparation, we compared a dilution series of HeLa PAC digest from 100 ng to 10 ng, thus starting from desalted peptides transferred to Evotips for injection, to a dilution series of intact cells sorted using cellenONE corresponding to 400, 200 and 40 HeLa cells, which were processed using One-Tip and thus only lysed in the Evotip. The protein amount in the cells corresponds to approximately 100, 50, and 10 ng of starting material, assuming 100% lysis efficiency and recovery. It is important to note that the number of cells sorted by the cellenONE represents the maximum number of cells potentially present in each sample, but the transfer likely resulted in cell loss. Nevertheless, this represents the fairest comparison of the two methods possible with our current instrumentation with a disadvantage for the One-Tip approach. We then compared the protein and peptide numbers identified in each sample (Fig. 2f, g). Overall, the protein and peptide numbers in samples from cellenONE were close to their 50% equivalent in peptide loading or slightly lower (i.e. 400 cells were close to 50 ng). The variation was higher in lower cell numbers, which is likely due to loss of cells during transfer to the tips. Finally, the percentage of missed cleavages was higher in One-Tip samples than in samples from PAC digestion (Fig. 2h).

## One-Tip enables single-cell proteome analysis of mouse pre-implantation embryos

To demonstrate the versatility and sensitivity of the One-Tip workflow, we next studied the development of mouse pre-implantation embryos at single-cell level from oocyte and zygote to 4-cell stage. The zona pellucida, the extracellular glycoproteinaceous coat of oocytes, zygotes, 2-cell and 4-cell stages, was chemically removed and single cells were dissociated under a microscope and dispensed individually into Evotips, pre-loaded with cell lysis and digestion buffer (Fig. 3a). This whole procedure was performed in one day for each batch of samples including the LC-MS/MS analysis. In total, 6216 protein groups were identified with a median of 5428, 5417, 5259, and 4779, for each developmental stage, respectively (Fig. 3b). In comparison to the very recent analysis of human pre-implantation embryos from Dang et al.[13] in which the authors employed a sophisticated sample preparation workflow and LC-MS gradients of 4 times longer duration than the one used in this study, we identify ~2300 more proteins and ~50,000 more peptides as well as ~3x peptides per protein on average (5.3 against 14.6), per stage (Supplementary Fig. 5a, b). Moreover, the variation in protein number, the overlap (Fig. 3b), and correlation of protein quantities between samples from a same condition was greatly improved (Fig. 3c, Supplementary Fig. 5c), demonstrating that the One-Tip workflow coupled to Evosep One and Orbitrap Astral is faster, more sensitive, more reproducible and has better quantification accuracy. Interestingly, the summed protein intensities in each sample were equal between oocytes and zygotes, while they were half those intensities in cells from the 2-cell stage and one-fourth in cells from the 4-cell stage, matching the reduction in cell volume during the two successive cleavages of the zygote (Fig. 3d). Although our analysis benefited from the use of the latest generation of mass spectrometer compared to an older generation in the study from Dang et al., which would inflate these numbers, our study highlights that using our

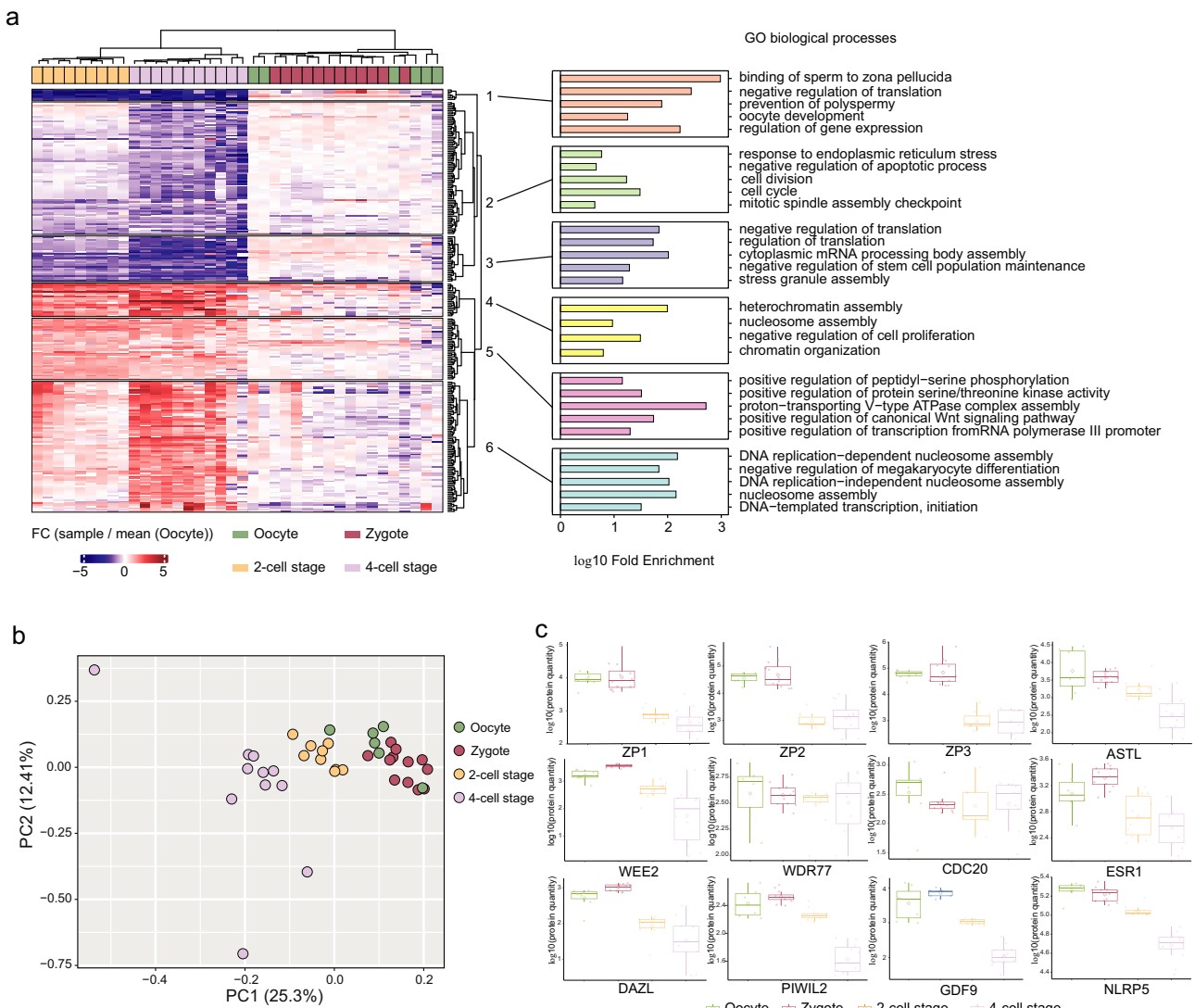

**Fig. 4 | Biological insights from the mouse pre-implantation embryos.**
**a** Unsupervised hierarchical clustering using canberra and ward.D2 methods of proteins that shows significant regulation (*p* < 0.05 from one-sided ANOVA) and FC > 2 fold in any of the developmental stages compared to the oocyte group (left). GO biological processes enrichment analysis of the protein clusters from the heatmap (right). Number of biological replicates: *n* = 6 for oocyte, *n* = 12 for zygote, *n* = 9 for 2-cell stage and *n* = 11 for 4-cell stage. **b** PCA of the normalized protein intensities for each samples of our study. Only proteins that were quantified in all samples were considered (*n* = 2953). **c** Abundances of twelve proteins related to reproductive processes across four developmental stages. For the boxplots, the lower and upper hinges correspond to the first and third quartiles. The upper whisker extends from the hinge to the largest value no further than 1.5 × IQR from the hinge (where IQR is the inter-quartile range). The lower whisker extends from the hinge to the smallest value at most 1.5 × IQR of the hinge. Number of biological replicates: *n* = 7 for oocyte, *n* = 12 for zygote, *n* = 10 for 2-cell stage and *n* = 12 for 4-cell stage. Source data are provided as a Source Data file.

minimalistic One-Tip workflow enables the acquisition of comprehensive datasets with greatly limited sample material.

Finally, we investigated whether biologically meaningful information can be retrieved from the mouse embryo cell analysis. We then performed unsupervised hierarchical clustering of differentially expressed proteins that were up or downregulated in any of the conditions compared to oocytes (Fig. 4a). Each sample group was separated on the heatmap[14] apart from the zygotes and oocytes as on the PCA. Gene ontology (GO) enrichment analysis of clusters reveals downregulation of proteins related to sperm binding to the zona pellucida, prevention of polyspermy, response to ER stress and upregulation of proteins related to DNA replication and heterochromatin assembly in 2-cell and 4-cell stages compared to oocytes, among other pathways. These results were globally in agreement with Dang et al. validating the quality of our analysis (Fig. 4a). The principal component analysis (PCA) of normalized protein intensities showed that oocyte and zygote group together and the 2-cell and 4-cell stages are

separated in a time-wise manner on PC1 (Fig. 4b). We then focused on the quantities of twelve proteins related to reproductive processes across four developmental stages (Fig. 4c). Notably, ZP3, a protein critical for zona pellucida formation[15], decreased during the 2-cell stage, aligning with the first cell division, but stabilized at the 4-cell stage, indicating dynamic changes in early embryonic development. In contrast, the quantities of WDR77, involved in transcriptional regulation[16], and CDC20, essential for cell division processes[17], remained stable across all stages. This consistency suggests continuous mitotic and transcriptional activity from the zygote stage, highlighting their crucial roles in early embryonic development.

**Scale One-Tip down to single HeLa cells with single cell dispenser**
Recently, the Uno Single Cell Dispenser™ (equivalent to the HP D100 Single Cell Dispenser) was described for single-cell sorting for proteomics analysis[18]. Since the design of the instrument allows

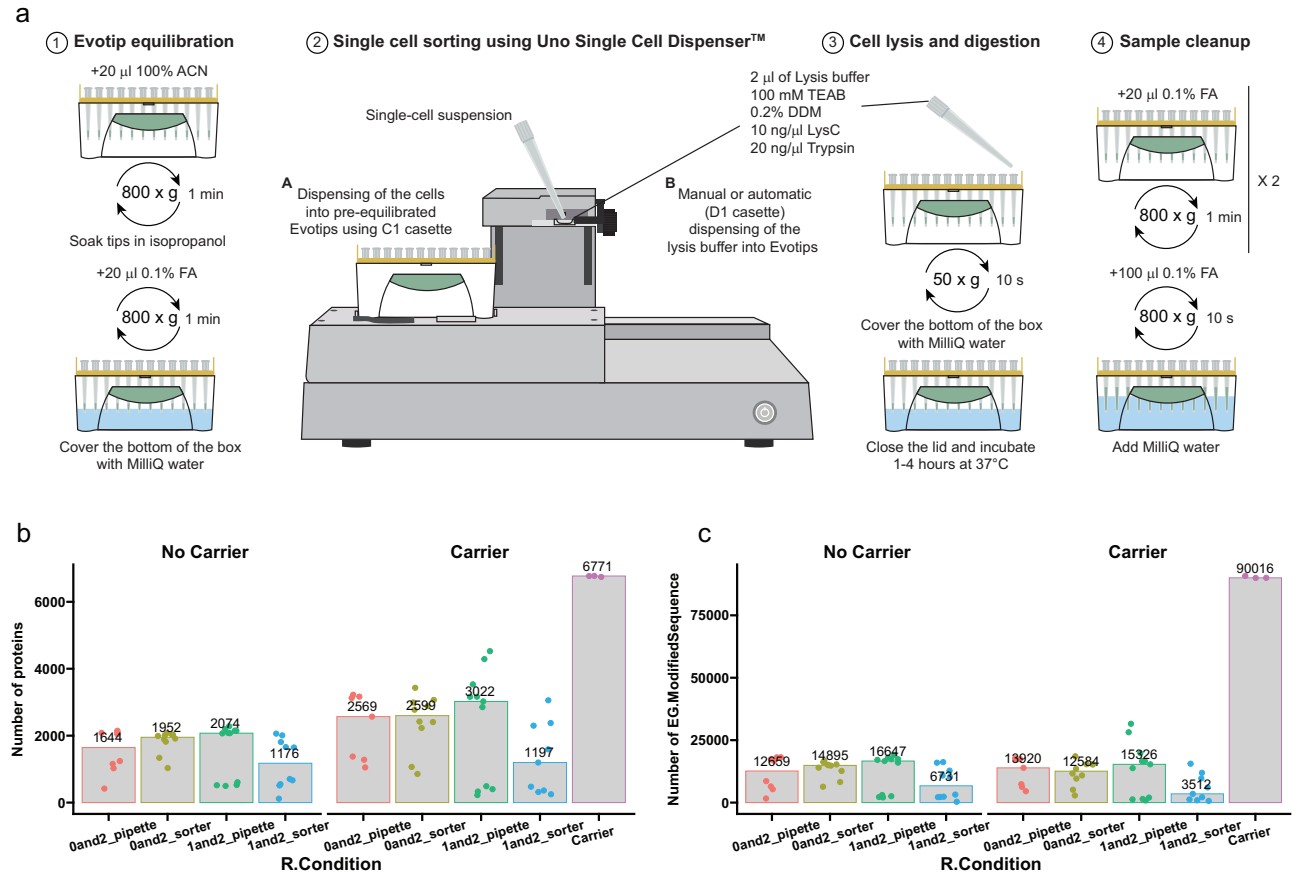

**Fig. 5 | Scale One-Tip down to single HeLa cells with single cell dispenser.**
**a** Workflow of single cell proteomics using One-Tip and Uno Single Cell Dispenser.
**b**, **c** Number of proteins (**b**) and peptides (**c**) identified in different sample preparation conditions. Number of samples: $n = 7$ 0and2_pipette, $n = 10$ 0and2_sorter, $n = 11$ 1and2_pipette, $n = 10$ 1and2_sorter, $n = 3$ carrier. Source data are provided as a Source Data file.

fitting a box of Evotips, we evaluated whether the One-Tip workflow could be amenable to single-cell analysis and tested multiple experimental set ups (Fig. 5a). In short, we either placed 1 µl of lysis and digestion buffer or nothing, sorted the cells and then dispensed 2 µl of Lysis buffer either manually (pipette) or with the dispenser. The results show that single-cell analysis is feasible using the Uno Single Cell Dispenser™ coupled to One-Tip with more than 2000 proteins quantified on average for the best condition (1 µl + 2 µl manually) with no carrier or reference sample[8,19] with higher signal to boost IDs during analysis with Spectronaut and more than 3000 with a carrier sample (Fig. 5b, c). Some of the tips had little signal suggesting that the cells did not reach the bottom of the Evotips and were not lysed and digested properly. However, consistent signal was detected in up to 80% of the samples, where single cells were supposed to be present according to the instrument software, which is lower than what was reported by Sanchez-Avila et al.[18] using a 384-well PCR plate. While the number of peptides and proteins identified and quantified are lower than for SCP analysis using the cellenONE instrument and the same LC-MS[8], using the combination of Uno Single Cell Dispenser™ and One-Tip is an economical and simple alternative to some of the more complicated and costly workflows used currently. Lastly, the workflow presented here has room for improvement both in terms of percentage of single cells recovered and in terms of peptide and protein numbers, and this should be further developed in the future.

**High-sensitivity workflow for extracellular vesicles with One-Tip**
To further showcase the broad applicability and sensitivity of the One-Tip workflow, we processed extracellular vesicles (EV) extracted from blood plasma rather than intact cells. We isolated EVs from the plasma of male donors as described by Kverneland et al.[20] and loaded a dilution series from 480 ng to 16 ng (estimated based on the protein concentration of plasma during EV purification) (Fig. 6a) on Evotips prepared for One-Tip analysis (Fig. 6b). We quantified more than 3000 proteins and 29000 peptides with 16 ng of starting material and close to 4500 proteins and 48000 peptides with 480 ng, the latter being on par with the state-of-the-art in plasma EV analysis[21] (Fig. 6c). Additionally, the number of protein and peptides identified scales with the amount of sample loaded, as observed with the mouse embryo data. Thus, this data shows that One-Tip can readily be applied to other types of samples than single cells or cell suspensions and will enable more sensitive, streamlined and higher throughput analysis of EVs. This is particularly relevant in the field of plasma analysis and biomarker discovery as well as in the EV field in general, where the amount of EVs originating from plasma or cells can be limited.

## Discussion
Taken together, our results demonstrate that the analytical depth achieved with the One-tip workflow is superior to standard workflows in the field of proteomics. Its simplicity, user-friendliness, and scalability render it highly suitable for a variety of experimental setups, ranging from single-cell analysis to bulk proteomics. This approach effectively eliminates the need for complex methodologies, specific equipment, and specialized proteomics expertise during the sample preparation phase. As a result, the One-Tip workflow substantially bridges the gap between proteomics and biology, simplifying LC-MS analyzes and making them more accessible and compatible with a

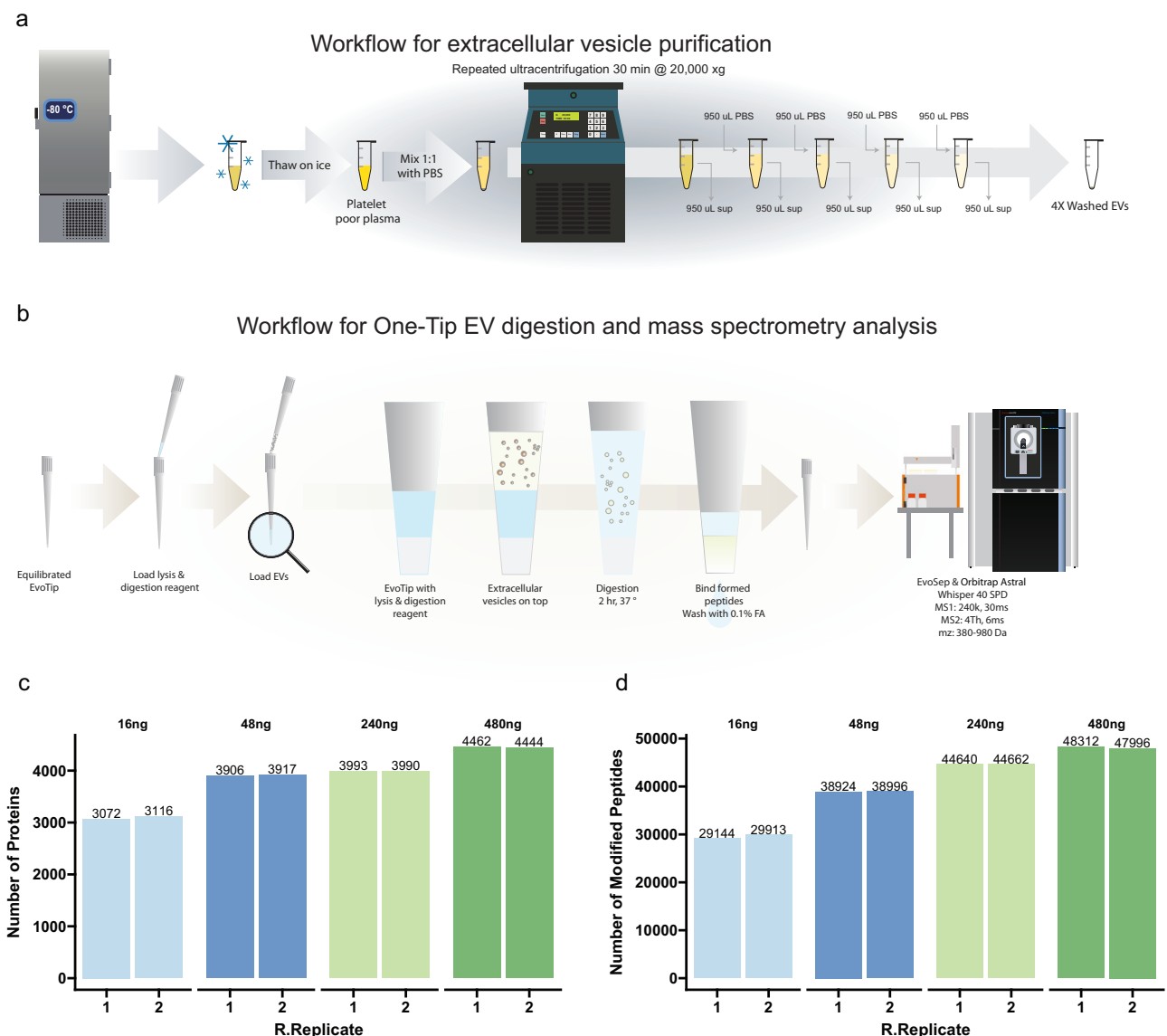

**Fig. 6 | One-Tip analysis of extracellular vesicles. a** Workflow for extracellular vesicle purification. **b** Workflow for One-Tip EV digestion and mass spectrometry analysis. **c**, **d** Number of proteins (**b**) and peptides (**c**) identified in different loading amounts. *N* = 2 technical replicates per condition. Source data are provided as a Source Data file.

broader range of biological applications. The workflow is adaptable to platforms such as cellenONE and Uno Single Cell Dispenser™, enhancing its versatility.

Additionally, One-Tip is capable of analyzing a diverse array of sample types, including FACS-sorted cells, human pre-implantation embryos, rare cell populations, and tissue slices. This is particularly relevant for smaller cell types, like T-cells, where lower number of proteins are expected to be identified as it correlates with the protein content (approximately 25 pg per cell). For such cells, single-cell analysis may offer limited biological insights, and the One-Tip workflow presents an effective alternative. Furthermore, we demonstrate the applicability of One-Tip to non-cellular samples, such as extracellular vesicles, showcasing its utility in a wide spectrum of research contexts. This adaptability makes One-Tip a valuable tool in the arsenal of modern proteomics, enabling a more inclusive approach to sample analysis and facilitating a deeper understanding of various biological systems.

## Methods

This study complies with all relevant ethical regulations. Animal work was conducted according to license no. 2021-15-0201-00851, approved

by the Danish National Animal Experiments Inspectorate, and performed according to national and local guidelines.

### HeLa cell lines

HeLa human cervix carcinoma cells (ATCC CCL-2) were cultured in DMEM (Gibco, Invitrogen), supplemented with 10% fetal bovine serum, 100 U/ml penicillin (Invitrogen), 100 µg/ml streptomycin (Invitrogen), at 37 °C, in a humidified incubator with 5% $CO_2$. At ~80% confluence, cells were detached using trypsin and washed twice with Phosphate Buffered Saline (PBS) from Gibco (Life Technologies), before being resuspended in degassed PBS.

### Detailed One-Tip sample preparation workflow

1. Determine HeLa cell concentration using a cell counter and dilute to the desired concentrations with PBS.
2. Prepare the Evotips following the vendor's instructions:
   - Rinse: Wash dry Evotips with 20 µl of Solvent B (0.1% FA in acetonitrile) and centrifuge at 800 g for 60 seconds.
   - Condition: Soak the Evotips in 100 µl of 1-propanol until they turn pale white.

- Equilibrate: Saturate the conditioned Evotips with 20 μl of Solvent A (0.1% FA in water) and centrifuge at 800 g for 60 seconds.

3. Pipette 5 μl of lysis and digestion buffer into the Evotips. The buffer contains 0.2% n-Dodecyl-β-D-Maltoside (DDM), 100 mM TEAB, 20 ng/μl Trypsin, and 10 ng/μl Lys-C.
4. Pipette 5 μl of cells into the Evotips.
5. Briefly centrifuge the Evotips at 50 g to mix the buffer and cells and prevent the formation of air bubbles.
6. Add water to the Evotip box to the level of the C18 resin in the Evotips.
7. Incubate the Evotip box at 37°C for 1 to 4 hours.
8. Continue with the vendor's instructions with a slight modification:
   - Load: add 50 μl of Solvent A to the Evotips and centrifuge for 60 seconds at 800 g.
   - Wash: Rinse the tips with 20 μl of Solvent A and centrifuge for 60 seconds at 800 g.
   - Wet: Add 100 μl of Solvent A to the tips and centrifuge for 10 seconds at 800 g to keep the tips wet.

### HeLa cell isolation with CellenONE

HeLa cells were diluted with degassed PBS to ~200 cells/μl. 20 cells were isolated into a 96 well-plate preloaded with 4 μl PBS using the CellenONE system. After isolation, samples were prepared following the One-Tip workflow.

### Isolation of mouse oocytes, zygotes, 2-cell and 4-cell stages

Animal work was conducted according to license no. 2021-15-0201-00851, approved by the Danish National Animal Experiments Inspectorate, and performed according to national and local guidelines. Mice were kept in designated rooms in individually ventilated cages at a temperature of 22 °C ( ± 2 °C), with a humidity of 55% ( ± 10%), air in the room was changed eight to ten times per hour and dark/light cycle is 12 h/12 h, light from 6 am to 6 pm. Mouse husbandry is performed according to Danish regulations for animal experiments.

Ovulation of prepubescent (4-week old) C57BL/6NRj females (mus musculus) was induced by intraperitoneal (IP) injection of PMSG (HOR-272, Prospec), 5IU/female, followed by IP injection of hCG (Chorulon Vet, Pharmacy), 5IU/female, 47 h later. After the second injection, females were set in breeding with C57BL/6NRj stud males. Next morning, the females were euthanized and oviducts were dissected to harvest the cumuli containing zygotes and unfertilized oocytes. Cumuli were disaggregated by 10-minute incubation in Hyaluronidase (H4272, Sigma-Aldrich). Zygotes were sorted out assessed by the presence of the second polar body and cultured in KSOM medium (MR-106-D, Merc Millipore). On the day of harvesting a group of both zygotes and oocytes were processed for analysis, while other zygotes were incubated in KSOM at 37 °C and 5% CO₂ for 24 h or 48 h. At the different developmental stages, oocytes and embryos were incubated in Tyrode´s Acidic Solution (T1788, Sigma-Aldrich) for 5 to 15 s in order to remove the zona pellucida. Naked oocyes and zygotes were washed in PBS (20012-027, Life Technologies) and individually transferred to Evotips using a 100 μm diameter glass capillary. 2-cell and 4-cell embryos were further disaggregated by 5-minute incubation in Ca2 + - and Mg2 + -free KSOM and pippeting with a 50 μm diameter glass capillary in PBS afterwards; individual blastomeres were then transferred to Evotips. The Evotips were preloaded with 2 μl of lysis and digestion buffer and 2 μl of PBS. Incubation time was 4 h. Number of biological replicates: n = 7 for oocyte, n = 12 for zygote, n = 10 for 2-cell stage and n = 12 for 4-cell stage. A total of 20 donor females and 8 stud males were used in these experiments.

### Cell isolation using the Uno Single Cell Dispenser™

Cell were prepared as for the sorting with the CellenOne and were passed through a 45 μm cutoff mesh to ensure that no clumps were present prior to sorting. Evotips were prepared for One-Tip analysis as described above and in Fig. 5a until the step where the lysis buffer is added. Since the lysis buffer is diluted 1:1 with the cell suspension in PBS in a regular One-Tip experiment, we pre-diluted the lysis buffer 1:1 with PBS. Then either 1 μl was added manually followed by centrifugation at 50 g for 10 s or no lysis buffer was added. After this, water was added to the Evotip box to cover only the tip of the Evotips and the box was placed on the Uno Single Cell Dispenser™ for single-cell dispensing. 20 μl of cell suspension were added to a C1 cassette and single cells were dispensed into the Evotips. After cell dispensing, 2 μl of diluted lysis buffer were added either by pipetting or with the dispenser using a D1 cassette. Water was removed and the tips were centrifuged at 50 g for 10 s so that the lysis buffer is in contact with the Evotip C18 phase. Water was added again to cover the bottom of the box and it was incubated at 37°C for 2 h. The rest of the sample preparation followed the One-Tip workflow. The number of sample was: n = 7 0and2_pipette, n = 10 0and2_sorter, n = 11 1and2_pipette, n = 10 1and2_sorter, n = 3 carrier.

### EV preparation

Venous blood samples were collected in citrate tubes (Vacuette, 3.2% sodium citrate, Greiner Bio-One) from male donors and centrifuged at 2,000 g for 10 min within 30 min after blood collection. The supernatant (platelet rich plasma) was transferred to a new tube and subject to a second centrifugation at 3000 g for 10 min to pellet remaining platelets and the platelet-poor supernatant was aliquoted, snap-frozen, and stored at −80°C. At the day of EV isolation, the platelet-poor plasma was thawed on wet ice and processed for EV enrichment as described in Fig. 6a. In short, 500 μl of plasma was diluted 1:1 with PBS pH 7.2 (Gibco cat# 20012-019) and centrifuged at 20,000 g for 30 min. 980 μl of supernatant were removed and 980 μl of PBS were added on top of the pellet (1:50 dilution), followed by centrifugation at 20,000 g for 30 min. This operation was performed 4 times in total and after the last centrifugation 980 μl of supernatant were removed and the remaining 20 μl containing the EV-pellet were further processed. The protein content in the EV-preparation was estimated based on the dilution of the total protein concentration in plasma (C = 60 g/L) and 4 ug of EVs in total (with a recovery of 90% after each centrifugation). The final EV preps after 4 centrifugations (3 washes) were diluted to obtain amounts equivalent to 16, 48, 240 and 480 ng in 4 μl. These 4 μl of each dilution in duplicates were added to Evotips prepared for One-Tip analysis containing 4 μl of lysis buffer and processed following the One-Tip protocol described above, prior to LC-MS/MS analysis. Experiment was performed in duplicate for each condition.

### PAC

Protein aggregation capture (PAC) was performed as previously described[11,22]. Briefly, HeLa cells were lysed in 5% SDS, 100 mM Tris pH 8.5, 5 mM tris(2-carboxyethyl)phosphine (TCEP), and 10 mm 2-chloroacetamide (CAA) and sonicated using a Branson probe sonicator 1 min using 3 s pulse and 30% amplitude. Protein concentration was measured using BCA assay, and 100 μg of proteins were used for PAC digestion. PAC digestion was conducted on a KingFisher Flex (Apex, Duo Prime) system (Thermo Fischer Scientific), sample volume was adjusted to 300 with lysis buffer, then 10 μl of MagReSyn® Hydroxyl beads (Resyn Biosciences) were added (2:1, beads/protein, w/w) followed by 700 μl of 100% acetonitrile, beads were rinsed 3 times with 100% acetonitrile, and 2 times with 70% ethanol and on-bead digestion was conducted o/n in 50 mM ammonium bicarbonate containing lysyl endopeptidase (1:500, enzyme/protein, w/w) and trypsin (1:250, enzyme/protein, w/w). Samples were acidified using formic acid to pH <3 and loaded onto Evotips prior to LC-MS/MS analysis.

### LC-MS/MS

LC-MS/MS analysis was performed on an Orbitrap Astral MS using Thermo Tune software (version 0.4 or higher) coupled to an Evosep

One system (EvoSep Biosystems). Samples were analyzed in 40SPD (31-min gradient) using a commercial analytical column (Aurora Elite TS, IonOpticks) interfaced online using an EASY-Spray™ source. The Orbitrap Astral MS was operated at a full MS resolution of 240,000 with a full scan range of 380 – 980 m/z when stated. The full MS AGC was set to 500%. MS/MS scans were recorded with 2Th isolation window, 3 ms maximum ion injection time (IIT) for HeLa samples and 4Th and 6 ms IIT for mouse embryonic samples, for the dilution series of PAC and CellenONE-sorted cells and for the single-cell analysis using the Uno dispenser. MS/MS scanning range was from 380–980 m/z were used. The isolated ions were fragmented using HCD with 27% NCE.

## MS data analysis

Raw files were analyzed in Spectronaut v18 (Biognosys) with a spectral library-free approach (directDIA + ) using the human protein reference database (Uniprot 2022 release, 20,588 sequences) for HeLa samples, and the mouse protein reference database (Uniprot 2022 release, 21,989 sequences) for the mouse embryo single cells complemented with common contaminants (246 sequences). Note, as the protocol does not involve reduction and alkylation, database searches were performed with free cysteine sulfhydryls and hence cysteine carbamylation was not set as a fixed modification, whereas methionine oxidation and protein N-termini acetylation were set as variable modifications. Precursor filtering was set to perform based on Q-values, and cross run normalization was checked. Each experiment with different number of cells was analyzed separately, and samples from different digestion times were searched with and without enabling method evaluation and indicating the different conditions (each one with $n = 3$ experimental replicates) in the condition setup tab.

## Reporting summary

Further information on research design is available in the Nature Portfolio Reporting Summary linked to this article.

## Data availability

The mass spectrometry proteomics data generated in this study have been deposited to ProteomeXchange Consortium (http://proteomecentral.proteomexchange.org) via the PRIDE partner repository with dataset identifier PXD044991. Source data are provided with this paper.

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

## Acknowledgements

Z.Y. was supported by the Non-profit Central Research Institute Fund of the Chinese Academy of Medical Sciences (grant no. 2023-RC180-03), the Chinese Academy of Medical Sciences (CAMS) Innovation Fund for Medical Sciences (2022-I2M-2-004, 2023-I2M-2-005) and the NCTIB Fund for the R&D Platform for Cell and Gene Therapy. Work at The Novo Nordisk Foundation Center for Protein Research (CPR) is funded in part by a donation from the Novo Nordisk Foundation (NNF14CC0001). N.B and J.V.O. received funding from the Innovation Fund Denmark under the grant ERA-PerMed-JTC2022-OVA-PDM for the project. P.S. is funded by the Swedish Research Council (2022-00323). A.E., O.Ø., and J.V.O.

received support from the European Research Council ERC Synergy Grant 810057-HighResCells.

## Author contributions

Z.Y., P.S., and J.V.O. conceptualized the study. Z.Y. and P.S. performed most of the experiments and data analysis. J. M.-G. performed the experiment on mouse embryos. A.E., D. B.-J., and N. B. contributed to the benchmarking experiments. M. L., Y. G., L. S., R. T., D. B.-J., and N. B. contributed to optimizing and performing the single-cell analysis using the Uno cell dispenser. O. Ø. performed the analysis of extracellular vesicles. J. X. helped with data analysis of the mouse embryo. Z.Y., P.S., and J.V.O. wrote the manuscript. All authors read, edited, and approved the final version of the manuscript.

## Competing interests

Yuan Guo, Lesley Schultz and Rafaela Truffer are employees of Tecan, the vendor of the Uno Single Cell Dispenser. Dorte B. Bekker-Jensen, Nicolai Bache are employees of Evosep Biosystems, manufacturer of the Evosep instrumentation and Evotip used in this work. Other authors declare no competing interests.
