## [Peer Review File · Nature Communications]

Reviewers' Comments:

Reviewer #1:

Remarks to the Author:

Ye, Sabatier and colleagues introduce a simple and quick sample preparation workflow for proteomics. This is important work of high interest that needs to be published, I look forward to using it in my lab.

Two extra benchmarks can help improve the paper further.

First, a comparison of the One-Tip workflow to direct loading of PAC-prepared bulk digest, diluted to the same cell number equivalent. This can be done, for example, down to 20-HeLa-cells-equivalent.

Second, a benchmark of the One-Tip workflow down to actual single HeLa cells, isolated with any suitable method (FACS, Tecan Uno, etc), not necessarily CellenONE, in comparison to bulk PAC digest diluted to the respective cell number-equivalent.

The above benchmarks will characterise the degree of losses introduced by One-Tip when dealing with low cell numbers. It is fine if they show inferior performance to a diluted PAC sample – this is expected. Benchmarks can also be carried out on any MS instrument, not necessarily Astral with the narrow-window DIA method.

How does the missed cleavage rate with One-Tip compare to that in PAC samples?

It's likely that significantly higher protein numbers can be obtained by analysing the runs from the whole dilution series together, in this case, run-specific protein group q-value filter needs to be set to 1%.

Would be helpful for the readers to also have the specific version of PAC described in Methods, in addition to being referenced.

One-Tip is referenced as One-Pot in Methods.

Vadim Demichev

Reviewer #2:

Remarks to the Author:

Thank you for the opportunity to read and review this early version of "One-Tip enables comprehensive proteome coverage in minimal cells and single zygotes".

Given the general excitement around low input proteomics today, I think that this will be well received. In addition, the study utilizes the new hybrid Orbitrap Time of Flight instrument and has some level of novelty based on the use of this hardware alone.

As it stands, it is my opinion this is likely a suitable home for this study, but some improvements on the data visualization and presentation side should be made prior to publication. The study as written could use some overall improvements in language here and there to make this seem a little less like a sales pitch for an expensive new instrument. While I think the chosen format has a limitation on the number of citations, I do think more than 8 are warranted in the main body of the text.

Minor comments:

Line 36 - I'm not sure "mere" is essential here. Other fast digestion methods do exist.

Line 48) I don't see where "finely tuned" is an accurate reflection of the mass analyzer as used here. Please adjust this language

Figure 1 is likely far too busy to convey well to print form. Also, it appears the default color and fill template in R/ggPlot was used and some minor changes will have large improvements. In particular the figure key in Figure 1D appears to contain the letter "A" within each block. I would recommend changing the color template to solid colors to hide this likely accidental icon. I'm not sure the grey fill default is the best for this figure either, but that may be personal taste.

Figure 1e may be better scaled to demonstrate the quality of overlap, but again, this may be a personal taste.

Lines 60-70 - I get it, you estimated cells by counting and that is probably just fine, I found Figure 1B and 1C confusing due to the inclusion of CellenOne counted single cells in the bar charts. I'm not entirely sure on a reread where this fits in the figure. I'll leave a note here, if the other reviewers are okay with cell counting and those relative errors, I won't be a jerk about it.

Lines 93-101; If you are going to highlight the Dang et al., study and the superiority of your new method, I think that you should also highlight that the instrument they used was now several generations behind the current models. Or this could be rephrased to seem less like a competitive statement.

Line 122, please elaborate on what a cranberry analysis is or provide a citation

Line 195, please clarify why different isolation window sizes were used for HeLa and for the mouse cells

Again, it is a nice study, I think it could use some polishing before publication.

Reviewer #3:

Remarks to the Author:

In the manuscript the authors present a method for preparing samples from small cell populations or few marginally large cells (zygotes). The manuscript contains some technical development with a proof-of-concept application; however, the minimal novelty and scientific content make it difficult to consider it a scientific publication.

- The major problem of streamlined methods for low-input or single-cell proteomics not being available is greatly overexaggerated. There are one-pot workflows developed by the labs of Karl Mechtler, Erwin Schoof and Ryan Kelly. Where the samples are directly prepared in a well-plate format and directly injected into the LC/MS with no transfers steps.

- The chosen reference approach for their method is highly questionable. It has been shown by two independent studies that transferring low-input sample by pipetting leads to drastic losses (Matzinger et al, 2023, Petrosius et al, 2023). The current comparison in a way reiterates the same finding. To truly gauge the performance of the One-tip method the following points should be addressed:

- o The samples seem to have been prepared at suboptimal conditions. The 20 cell with CellenOne are prepared in a much greater volume (5uL), than the standard (<1uL), which has been well documented to affect the obtained proteome coverage from low input samples.

- o The performance should be compared to direct injection methods developed by other labs where no transfers steps are needed (e.g. Matzinger et al, Liang et al, 2021)

- Authors completely overlook the existence of FACS, which has been successfully used for low-input proteomics and is generally easily accessible in most institutes. Furthermore, the Uno Single Cell Dispenser™ from Tecan, that can carry out all the necessary cell isolation and reagent dispensing should be consider, as it can prepare samples for direct injection in a well-plate format that is compatible with routine LC systems, such as the Easy-nLC, Vanquish Neo and nanoElute.

- The pipetting method for cell isolation is likely to suffer from inaccuracies. Assuming the cell aliquoting process follows a poisson sampling distribution the 95% confidence interval is $\text{aliquoted_cell_number} \pm \sqrt{\text{aliquoted_cell_number}} * 2$, meaning that the authors are expected to have used from 14 to 26 cells for the samples when 20 cells was the aim. Why they didn't take actual cell counts from the CellenOne seems strange and should be addressed.

- The findings from oocyte-zygote experiments are purely technical – it would have been nice to see a bit more biological interpretation to better understand why these experiments were undertaken in the first place.

- The authors use a state-of-the art analytical column (Aurora Elite TS) and the arguably most advanced MS instrument currently available, however the numbers are compared to a study that used a home packed column and a previous generation instruments timsTOF Pro. It is impossible to judge whether the improved proteome is a result of the One-Tip sample preparation or simply much more advanced instrumentation. This discussion needs to be revised to make this much

more clear, or their experimental workflow directly compared to a previous generation MS instrument to be able to assess the impact of the one-pot workflow.

- The workflow is tied to the EvoSep One LC platform and will not be applicable to user of other vendor instruments.

Minor:

- The previous research in the field is poorly referenced, making it unclear to less well-informed readers what the context of this work is and what the true advances actually are.

- The authors refer to their MS acquisition method as nDIA, however similar window sizes with the Orbitrap ASTRAL MS have already been published by the MacCoss lab under the standard DIA flag, so this reviewer doubts whether a new acronym is actually warranted.

- Some discussion where 20-3000 cell sample proteomics would be applicable in practice would be useful.

RESPONSE TO REVIEWER COMMENTS

NCOMMS-23-43900-T

Please find our point-by-point responses to the reviewer comments below in **blue text**.

Reviewer #1 (Remarks to the Author):

Ye, Sabatier and colleagues introduce a simple and quick sample preparation workflow for proteomics. This is important work of high interest that needs to be published, I look forward to using it in my lab.

Response : We thank you for your nice comments.

Query 1: Two extra benchmarks can help improve the paper further.

First, a comparison of the One-Tip workflow to direct loading of PAC-prepared bulk digest, diluted to the same cell number equivalent. This can be done, for example, down to 20-HeLa-cells-equivalent. Second, a benchmark of the One-Tip workflow down to actual single HeLa cells, isolated with any suitable method (FACS, Tecan Uno, etc), not necessarily CellenONE, in comparison to bulk PAC digest diluted to the respective cell number-equivalent.

The above benchmarks will characterise the degree of losses introduced by One-Tip when dealing with low cell numbers. It is fine if they show inferior performance to a diluted PAC sample – this is expected. Benchmarks can also be carried out on any MS instrument, not necessarily Astral with the narrow-window DIA method.

Response 1:

Response: We added a dilution of HeLa digest from PAC ranging from 100 ng PAC digest to 10 ng compared to 400 to 40 HeLa cells sorted with cellenONE and transferred onto Evotips loaded with lysis buffer (Fig. 2f, 2g). For the single cell analysis we used the Uno Single Cell Dispenser™ (Fig. 5). We added additional sections in the text accordingly. The reason for using 2 different sorters is that we are likely losing cells during the transfer from the cellenONE chip onto the Evotip and it is not possible to dispense more than one cell with the Uno Single Cell Dispenser™ at the moment. Therefore we only went down to 40 cells (10 ng of proteins in theory) with the cellenONE and true single cell with the Uno Single Cell Dispenser™.

Query 2: How does the missed cleavage rate with One-Tip compare to that in PAC samples?

Response: Thank you for your suggestion. We have discussed the “carrier effect” in our SCP manuscript as it is more relevant in that context. Here, for dilution series comparing PAC to cellenONE-sorted cells, we used “method evaluation” mode from SN for each different group so the protein number is not affected by other groups with different cell number or peptide loading, offering more insightful analysis of sample quality.

Query 3: It’s likely that significantly higher protein numbers can be obtained by analysing the runs from the whole dilution series together, in this case, run-specific protein group q-value filter needs to be set to 1%.

Response: We agree that analyzing the runs from the whole dilution series together will for sure lead to significantly higher protein numbers. However, for the dilution series comparing PAC to cellenONE-sorted cells, we used the “method evaluation” mode from SN for each different group

separately such that the protein number are not affected by other groups with different cell amount or peptide loading.

Query 4: Would be helpful for the readers to also have the specific version of PAC described in Methods, in addition to being referenced.

Response: We added a description of the PAC protocol used in this study.

Query 5: One-Tip is referenced as One-Pot in Methods.

Response: We changed the text.

Vadim Demichev

Reviewer #2 (Remarks to the Author):

Thank you for the opportunity to read and review this early version of "One-Tip enables comprehensive proteome coverage in minimal cells and single zygotes".

Given the general excitement around low input proteomics today, I think that this will be well received. In addition, the study utilizes the new hybrid Orbitrap Time of Flight instrument and has some level of novelty based on the use of this hardware alone.

As it stands, it is my opinion this is likely a suitable home for this study, but some improvements on the data visualization and presentation side should be made prior to publication. The study as written could use some overall improvements in language here and there to make this seem a little less like a sales pitch for an expensive new instrument. While I think the chosen format has a limitation on the number of citations, I do think more than 8 are warranted in the main body of the text.

We thank reviewer 2 for his or her nice words and agree that the article needs to be polished prior to publication. Consequently, we have now rewritten the manuscript and expanded the text and references accordingly.

Minor comments:

1) Line 36 - I'm not sure "mere" is essential here. Other fast digestion methods do exist.

Response: We removed it.

2) Line 48) I don't see where "finely tuned" is an accurate reflection of the mass analyzer as used here. Please adjust this language

Response: We adjusted it.

3) Figure 1 is likely far too busy to convey well to print form. Also, it appears the default color and fill template in R/ggPlot was used and some minor changes will have large improvements. In particular the figure key in Figure 1D appears to contain the letter "A" within each block. I would recommend changing the color template to solid colors to hide this likely accidental icon. I'm not sure the grey fill

default is the best for this figure either, but that may be personal taste.

Response: We thank the reviewer for pointing this out. Following the reviewer's advice to reduce the busy figure, we now split Figure 1 into 2 figures (now Figure 1 and Figure 2).

4) Figure 1e may be better scaled to demonstrate the quality of overlap, but again, this may be a personal taste.

Response: We adjusted the figure.

5) Lines 60-70 - I get it, you estimated cells by counting and that is probably just fine, I found Figure 1B and 1C confusing due to the inclusion of CellenOne counted single cells in the bar charts. I'm not entirely sure on a reread where this fits in the figure. I'll leave a note here, if the other reviewers are okay with cell counting and those relative errors, I won't be a jerk about it.

Response: We included a new analysis with cells accurately counted with the cellenONE instrument (Fig. 2f, 2g).

6) Lines 93-101; If you are going to highlight the Dang et al., study and the superiority of your new method, I think that you should also highlight that the instrument they used was now several generations behind the current models. Or this could be rephrased to seem less like a competitive statement.

Response: We agree that the comparison is not totally fair and we adjusted the text to the readers aware of it. We added the following paragraph in the manuscript:

“Although our analysis benefitted from the use of the latest generation of mass spectrometer compared to an older generation in the study from Dang et al., which would inflate these numbers, our study highlights that using our minimalistic One-Tip workflow enables the acquisition of comprehensive datasets with greatly limited sample material.”

7) Line 122, please elaborate on what a cranberry analysis is or provide a citation

Response: We assumed that reviewer 2 meant Canberra and we provided a citation in the text describing the mathematical formula DOI: [10.1007/978-3-319-63315-2_63](https://doi.org/10.1007/978-3-319-63315-2_63).

8) Line 195, please clarify why different isolation window sizes were used for HeLa and for the mouse cells

Response: We used different quadrupole isolation window sizes for the DIA because the amount of sample loaded with up to 1000 HeLa cells allowed us to use 2 Th isolation windows for higher specificity at the cost of sensitivity, and we chose to stay consistent in the whole dilution series analysis. We used a 4 Th window size for analyses with lower amount of samples including the single-cell analysis with the Uno dispenser to increase sensitivity as showcased in our recent SCP preprint <https://www.biorxiv.org/content/10.1101/2023.11.27.568953v1.abstract>.

Again, it is a nice study, I think it could use some polishing before publication.

Response : We thank you for your nice comments.

Reviewer #3 (Remarks to the Author):

In the manuscript the authors present a method for preparing samples from small cell populations or few marginally large cells (zygotes). The manuscript contains some technical development with a proof-of-concept application; however, the minimal novelty and scientific content make it difficult to consider it a scientific publication.

Query 1: The major problem of streamlined methods for low-input or single-cell proteomics not being available is greatly overexaggerated. There are one-pot workflows developed by the labs of Karl Mechtler, Erwin Schoof and Ryan Kelly. Where the samples are directly prepared in a well-plate format and directly injected into the LC/MS with no transfers steps.

Response 1:

We are sorry that there seems to be some confusion regarding our approach. Our One-Tip method was not designed specifically for single-cell proteomics. Rather, its primary advantage lies in its ability to deliver comprehensive proteome coverage, exemplified by identifying over 9000 proteins reproducibly from just 1000 cells and a few hours of sample preparation with minimal pipetting steps. These features position One-Tip as a valuable option for bulk proteomics, offering a proteome depth comparable to traditional methods like PAC in a much shorter time frame and much lower sample input.

While it is certainly true that streamlined methods developed by Karl Mechtler, Erwin Schoof, and Ryan Kelly enable direct well-plate format preparation and injection into LC/MS systems, these methods were specifically applied for single-cell proteomics. Moreover, a significant limitation of these methods, is the lack of a sample cleanup step. This can potentially compromise the sample quality and impact longevity of LC-MS systems.

To address these points and the feedback received from the reviewer, we have made several updates in our revised manuscript:

- 1) The Introduction has been extensively revised to adequately describe and reference the mentioned studies. We have also made it clear that *“our study introduces the One-Tip workflow, which is meticulously designed to overcome the prevalent issues in bulk proteomics.”*
- 2) We've added a new section demonstrating the application of our One-Tip method using Uno Single Cell Dispenser™ for actual single-cell proteomics.
- 3) We've expanded on other applications of One-Tip, including its use in plasma EV samples.

Query 2: The chosen reference approach for their method is highly questionable. It has been shown by two independent studies that transferring low-input sample by pipetting leads to drastic losses (Matzinger et al, 2023, Petrosius et al, 2023). The current comparison in a way reiterates the same

finding. To truly gauge the performance of the One-tip method the following points should be addressed:

The samples seem to have been prepared at suboptimal conditions. The 20 cell with CellenOne are prepared in a much greater volume (5uL), than the standard (<1uL), which has been well documented to affect the obtained proteome coverage from low input samples.

Response 2:

We thank the reviewer for the suggestions, which we have followed and now included the data with different volumes in Figure 1B.

Importantly, our research has demonstrated that the previous generation of Orbitrap mass spectrometer, the Orbitrap Exploris 480, can also yield high-quality results with samples prepared using the One-Tip method. This illustrates the broad applicability and versatility of the One-Tip method across different platforms of mass spectrometry technology.

Please also refer to Response 1 for the discussions.

Query 3: The performance should be compared to direct injection methods developed by other labs where no transfers steps are needed (e.g. Matzinger et al, Liang et al, 2021)

Response 3:

We believe it is important to recognize that the suggested references to the direct injection methods are primarily suited for single-cell analysis or very low sample inputs. Routine bulk analysis using direct injection is generally avoided in labs to maintain the longevity of LC columns and to preserve the cleanliness of the mass spectrometer.

To effectively evaluate the performance of our single cell proteomics data, we suggest the reviewer refer to our recent SCP preprint (<https://www.biorxiv.org/content/10.1101/2023.11.27.568953v1.abstract>). In this publication, we demonstrate the identification of over 5000 proteins and more than 40,000 peptides. These findings

significantly surpass the results of any other studies in the field of single cell proteomics to date.

Query 4: Authors completely overlook the existence of FACS, which has been successfully used for low-input proteomics and is generally easily accessible in most institutes. Furthermore, the Uno Single Cell Dispenser™ from Tecan, that can carry out all the necessary cell isolation and reagent dispensing should be considered, as it can prepare samples for direct injection in a well-plate format that is compatible with routine LC systems, such as the Easy-nLC, Vanquish Neo and nanoElute.

Response 4:

We thank the reviewer for pointing this out. Our decision to omit FACS was deliberate, as our experiments did not incorporate this technique, and we aimed to avoid overstating the scope of our method. However, recognizing its relevance, we have now included a discussion on FACS in our revised manuscript.

Moreover, regarding the Uno Single Cell Dispenser™ from Tecan, it is indeed an intriguing piece of equipment, particularly for single-cell analysis. However, its limitation of dispensing up to only 400 cells per chip and to not be able to dispense more than 1 cell per well/tip would have posed a significant challenge for the scale of our study. This limitation is a key factor that influenced our experimental design and methodology. That being said, we have followed the advice of the reviewer and conducted new experiments in which single cells were sorted using the Uno Single Cell Dispenser™, demonstrating the compatibility of this approach with the One-Tip method. This new data, included in our revised manuscript, showcases the potential of integrating cell sorting technologies with the One-Tip workflow, expanding its applicability in proteomic research.

Query 5: The pipetting method for cell isolation is likely to suffer from inaccuracies. Assuming the cell aliquoting process follows a poisson sampling distribution the 95% confidence interval is $\text{aliquoted_cell_number} \pm \sqrt{\text{aliquoted_cell_number}} * 2$, meaning that the authors are expected to have used from 14 to 26 cells for the samples when 20 cells was the aim. Why they didn't take actual cell counts from the CellenOne seems strange and should be addressed.

Response 5: We agree that the cell counting may be inaccurate. Consequently, we have followed the advice of the reviewer, and sorted 400, 200 and 40 cells with the cellenONE to also have a comparison with equivalent peptide load from PAC digest (Fig. 2f, 2g).

Query 6: The findings from oocyte-zygote experiments are purely technical – it would have been nice to see a bit more biological interpretation to better understand why these experiments were undertaken in the first place.

Response 6:

Thank you for the suggestion. We have included a new Figure 4c to describe the biological findings in our datasets in more depth.

The main reason for conducting the oocyte-zygote experiments was to address the technical challenges associated with sorting these samples using traditional methods. Given the small number of cells available and the risk of cellular damage due to excessive manipulation, traditional sorting technologies are not feasible for these samples. Our method highlights a simpler solution to this issue. The significance of our work lies in demonstrating a robust methodology that can potentially

be applied to human pre-implantation embryos. This opens new possibilities for in-depth studies in a field where sample handling and preservation are critical and challenging.

Query 7: The authors use a state-of-the-art analytical column (Aurora Elite TS) and the arguably most advanced MS instrument currently available, however the numbers are compared to a study that used a home packed column and a previous generation instruments timsTOF Pro. It is impossible to judge whether the improved proteome is a result of the One-Tip sample preparation or simply much more advanced instrumentation. This discussion needs to be revised to make this much more clear, or their experimental workflow directly compared to a previous generation MS instrument to be able to assess the impact of the one-pot workflow.

Response 7:

Thanks for the suggestions. We have followed the reviewer suggestion and implemented the following revisions to our study:

- 1) We have now included results obtained using a previous generation mass spectrometer, the Orbitrap Exploris 480 MS. This addition highlights the effectiveness and adaptability of the One-Tip method across various mass spectrometry platforms, reinforcing its broad utility. See also Response 2.
- 2) Regarding the enhanced proteome coverage compared to that achieved using the timsTOF Pro, it is important to note that the timsTOF Pro is capable of producing deep proteome coverage in 2-hour gradient runs. Although a systematic comparison between the timsTOF Pro and the Orbitrap Astral is yet to be conducted, we believe that the performance difference is not as significant as an approximate 20-fold increase by achieving five times more peptides in one-quarter of the acquisition time with the Orbitrap Astral. More crucially, as indicated in Supplementary Figure 1a of the Dang et al. study and our Supplementary Figure 5c, our results demonstrate a significantly better correlation (0.9 versus 0.7) in protein abundances across different samples. This marked improvement is largely independent of the mass spectrometry platform used, which underscores the superiority of our One-Tip workflow. We have clarified these points in our revised manuscript to better illustrate the impact of the One-Tip method relative to the advancements in mass spectrometry instrumentation.

Left: Supplementary Figure 1a of the Dang et al; Right: Supplementary Figure 5c of our study.

Query 8: The workflow is tied to the EvoSep One LC platform and will not be applicable to user of other vendor instruments.

Response 8:

We acknowledge that the One-Tip workflow is primarily designed for use with the Evosep One LC platform, primarily due to the integration of Evotips, a key feature of the Evosep One system. However, it's important to note that the Evosep One is compatible with mass spectrometers from leading manufacturers, including Thermo Fisher Scientific, Bruker, and AB Sciex. Given the widespread adoption of Evosep One systems among LC-MS users, this should not significantly limit the method's broad applicability.

Furthermore, there is an alternative approach for users of other LC systems. Samples prepared using the One-Tip method can simply be eluted into a well plate and then injected into a different LC system. This approach is similar to pre-cleaning samples as done in other one-pot methods, ensuring versatility. We have included this information in the Discussion section of our manuscript to clarify the adaptability of the One-Tip method across different LC platforms.

Minor:

- The previous research in the field is poorly referenced, making it unclear to less well-informed readers what the context of this work is and what the true advances actually are.

Response 9:

We rewrote the Introduction and added more references.

- The authors refer to their MS acquisition method as nDIA, however similar window sizes with the Orbitrap ASTRAL MS have already been published by the MacCoss lab under the standard DIA flag, so this reviewer doubts whether a new acronym is actually warranted.

Response10:

The new acronym was used in our recent article "Narrow-window DIA: Ultra-fast quantitative analysis of comprehensive proteomes with high sequencing depth" (bioRxiv 2023.06.02.543374; doi: <https://doi.org/10.1101/2023.06.02.543374>), which has now accepted for publication in Nature Biotechnology (<https://doi.org/10.1038/s41587-023-02099-7>). The new acronym was well accepted by the reviewers here.

- Some discussion where 20-3000 cell sample proteomics would be applicable in practice would be useful.

Response 11:

We agree with the reviewer. Consequently, we have updated the discussion section accordingly, and included a paragraph discussing One-Tip in context of FACS-sorted cells, human pre-implantation embryos, rare cell populations, tissue slices potentially. Particularly for smaller cells were the number of protein identified scales with protein contents such as T cells which contain around 25 pg per cell. For these cells, single cell analysis makes little sense and it would be hard to recover any biological insight from such SC datasets. We also demonstrate that One-Tip can be applicable to samples other than cells such as extracellular vesicles. We added this part to the discussion.

Reviewers' Comments:

Reviewer #1:

Remarks to the Author:

The authors have incorporated my suggestions.

I would further recommend to clarify why 200 cells on Figure 1c show higher protein numbers than 100ng PAC on Figure 2f, but this will not need to be reviewed.

Reviewer #3:

Remarks to the Author:

I would like to begin by commending the authors for significantly expanding the scope of the study, making it extremely difficult to deny the utility of the One-Tip workflow. While I am still not completely convinced by the stated superiority of it, this manuscript should not be held hostage by one man's skepticism and that judgment should be left to the general scientific community who will hopefully try out this approach. I would appreciate some general tempering of the language e.g. calling the workflow "ultra-sensitive by nature" when the single-cell results are markedly inferior to other approaches is not warranted. The authors have addressed all my initial concerns, but one and I would just have minor comments/clarifications:

Q1-R1 (and Q3-R3) Apologies on my part as I think my response was not verbose enough to properly express my concern. My focus was on the lowest 20 cell samples that was used as a reference prepared with the CellenOne. Such cell amounts have been used by both Ryan Kelly and Karl Mechtler without cleanup steps and can be directly injected (<https://www.nature.com/articles/s41467-024-45391-z#Sec11>, <https://pubs.acs.org/doi/full/10.1021/acs.analchem.0c04240>). The high preparation volume and subsequent transfer potentially drastically downgraded the quality of this reference sample. These issues have already partially been addressed in the main text. However, since 1000 cells are the true target ranges of the workflow direct injection should definitely not be attempted here.

Q2-R2 My primary concern was that the 20 cell preparation was carried out in too large a volume (5 μ L) and switching to 1 μ L would have probably improved the proteome coverage due to improved digestion kinetics. I am a bit confused what point preparing larger amount of cells in larger volume addressed here, but it is definitely not the concern I have raised. But again, if the target is thousands of cells this comparison is secondary and simply mentioning that a larger preparation volume was used which could negatively affect the protein number obtained with the CellenOne preparation.

New minor:

Use of the term "Carrier" to define the 3000 cell runs used to generate a library/co-search the single-cells with could be confusing considering that in TMT based workflows (SCoPE-MS) the carrier is defined as 100-200 sample that is injected together with the single-cells into the instrument.

RESPONSE TO REVIEWER COMMENTS

NCOMMS-23-43900-A

Please find our point-by-point responses to the reviewer comments below in blue text.

Reviewer #1 (Remarks to the Author):

The authors have incorporated my suggestions.

I would further recommend to clarify why 200 cells on Figure 1c show higher protein numbers than 100ng PAC on Figure 2f, but this will not need to be reviewed.

Response: The difference is marginal and this could be due to two reasons: 1) the 200-cell samples had 12 samples in total in the search, while the 100ng samples had only 4. As the reviewer stated in the last revision, this will change the numbers that can be identified in Spectronaut; 2) As we also mentioned in the manuscript, “these cell quantities were based on cell counting estimations, and therefore are approximations”, and this could lead to the marginal differences in the two different experiments.

Reviewer #3 (Remarks to the Author):

I would like to begin by commending the authors for significantly expanding the scope of the study, making it extremely difficult to deny the utility of the One-Tip workflow. While I am still not completely convinced by the stated superiority of it, this manuscript should not be held hostage by one man's skepticism and that judgment should be left to the general scientific community who will hopefully try out this approach. I would appreciate some general tempering of the language e.g. calling the workflow "ultra-sensitive by nature" when the single-cell results are markedly inferior to other approaches is not warranted. The authors have addressed all my initial concerns, but one and I would just have minor comments/clarifications:

Q1-R1 (and Q3-R3) Apologies on my part as I think my response was not verbose enough to properly express my concern. My focus was on the lowest 20 cell samples that was used as a reference prepared with the CellenOne. Such cell amounts have been used by both Ryan Kelly and Karl Mechtler without cleanup steps and can be directly injected (<https://www.nature.com/articles/s41467-024-45391-z#Sec11>, <https://pubs.acs.org/doi/full/10.1021/acs.analchem.0c04240>). The high preparation volume and subsequent transfer potentially drastically downgraded the quality of this reference sample. These issues have already partially been addressed in the main text. However, since 1000 cells are the true target ranges of the workflow direct injection should definitely not be attempted here.

Q2-R2 My primary concern was that the 20 cell preparation was carried out in too large a volume (5uL) and switching to 1uL would have probably improved the proteome coverage due to improved digestion kinetics. I am a bit confused what point preparing larger amount of cells in larger volume addressed here, but it is definitely not the concern I have raised. But again, if the target is thousands

of cells this comparison is secondary and simply mentioning that a larger preparation volume was used which could negatively affect the protein number obtain with the CellenOne preparation.

Response: We thank the reviewer for the kind words and comments. We agree that the 5 μ l volume was not the best condition for 20 cells. To make it clear, we added the following sentence to the manuscript:

“And the volume of 5 μ l master mix in this experiment might not be optimal for the different number of cells.”

New minor:

Use of the term “Carrier” to define the 3000 cell runs used to generate a library/co-search the single-cells with could be confusing considering that in TMT based workflows (SCoPE-MS) the carrier is the defined as 100-200 sample that is injected together with the single-cells into the instrument.

Response: This word was intentionally chosen, drawing from the TMT workflow analogy, to conceptualize "carrier proteome effects" within single-cell proteomics This effect was discussed in another study from our laboratory last year (<https://www.biorxiv.org/content/10.1101/2023.11.27.568953v1.abstract>), and it is now being adopted by other experts in the field (<https://www.biorxiv.org/content/10.1101/2024.01.20.576369v1.abstract>). To make it clear, we now cited these papers in the manuscript.